# Towards Flash Thinking via Decoupled Advantage Policy Optimization

## Abstract

Recent Large Reasoning Models (LRMs) have achieved remarkable performance in solving complex problems via supervised fine-tuning (SFT) and reinforcement learning (RL). Although existing RL algorithms significantly enhance model accuracy, they still suffer from excessively lengthy responses and overthinking issues, resulting in increased inference latency and computational consumption, especially for simple tasks that require minimal reasoning. To address this, we propose a novel RL framework, DEPO, to reduce inefficient reasoning for models. Our method mainly consists of three core components: (1) an innovative advantage decoupled algorithm to guide model reduction of inefficient tokens; (2) a difficulty-aware length penalty to lower the overall length of model responses; (3) an advantage clipping method to prevent bias in policy optimization. In our experiments, applied to DeepSeek-Distill-Qwen-7B and DeepSeek-Distill-Qwen-1.5B as base models, DEPO achieves a significant reduction in sequence length by 39% and reduces excessive reasoning paths in inefficient tokens, while outperforming the base model in overall accuracy.

## 1 Introduction

Recent large reasoning models (LRMs) (OpenAI, 2024; DeepSeek-AI et al., 2025) have achieved significant advances in mathematical reasoning and programming by leveraging extensive chains of thought (CoT) (Wei et al., 2023). These models emulate human-like deep thinking through mechanisms such as self-reflection, error correction, and the exploration of multiple solution strategies. However, chains of thought often contain long and redundant reasoning trajectories, a phenomenon known as the OverThink (Cuadron et al., 2025), which leads to substantial inference latency and high computational costs. For instance, models may repeatedly verify an already correct answer through redundant self-reflection or unnecessarily complicate simple problems by generating overly elaborate reasoning steps.

To address the overthinking problem, recent approaches can be broadly categorized into three directions. First, some methods construct preference datasets based on output length (Shen et al., 2025) for model training. However, this strategy suffers from preference mismatch and is labor-intensive for data construction. Second, other approaches incorporate a length penalty into the reward function to encourage more concise generation (Zhang & Zuo, 2025). While effective in reducing response length, these methods treat the entire model response as a whole and thus fail to guide the model in identifying and suppressing specific redundant reasoning segments. Moreover, the length penalty can distort the advantage estimation of individual tokens, leading policy updates in the wrong direction and ultimately degrading model accuracy. Third, recent work by Cheng et al. (2025) attempts to mitigate overthinking by extracting valid thinking tokens from the CoT and down-weighting the advantage values of invalid ones. However, this method relies solely on the length ratio between valid and invalid reasoning segments to modulate advantages, without accounting for the specific degree of overthinking in inefficient segments. Consequently, the model struggles to learn how to effectively suppress specific overthinking patterns.

Therefore, building upon the insights and limitations from these works, we propose to partition the model responses into efficient and inefficient segments, enabling us to mitigate overthinking in the inefficient parts while simultaneously reducing overall response length, bypassing the need for labor-consuming preference dataset construction. To achieve this, we propose **DE**coupled Ad-

**Figure 1: Illustration of proposed DEPO. DEPO enables token-level advantage estimation to update $\pi_{old}$ to $\pi_{\theta}$, in contrast to sequence-level methods in GRPO.**

vantage **P**olicy **O**ptimization (**DEPO**), an innovative RL algorithm that introduces three key innovations: (1) we introduce a decoupled advantage computation method for inefficient tokens and down-weight their gradient updates according to the degree of overthinking in the corresponding segment; (2) we implement a difficulty-aware length penalty that encourages shorter responses for easier questions while reducing overall output length; (3) we further propose an advantage clipping strategy to prevent reward fluctuations from steering policy updates in the wrong direction.

Our experiments demonstrate that DEPO effectively mitigates overthinking in inefficient reasoning segments, reducing redundant reasoning steps by more than 50%. Across multiple test sets, DEPO consistently shortens model responses while preserving or even slightly improving task accuracy compared to the base model. Specifically, when applied to the DEEPSEEK-DISTILL-QWEN-7B and DEEPSEEK-DISTILL-QWEN-1.5B models, DEPO achieves an average accuracy gain of 2.0% over the base model, accompanied by a 38.7% reduction in response length for DEEPSEEK-DISTILL-QWEN-7B and a 39.1% reduction for DEEPSEEK-DISTILL-QWEN-1.5B. These impressive results indicate that targeting redundant reasoning in inefficient responses can effectively mitigate overthinking in LRMs while preserving training accuracy.

In conclusion, our contributions can be summarized as follows:

- We propose a novel algorithm that decouples advantage computation for efficient and inefficient reasoning segments. By leveraging a pretrained GRM, our method precisely identifies the first reasoning step that leads to the correct answer, enabling an explicit separation of reasoning trajectories.

- We analyze the semantic characteristics of base models during the reasoning process and identify generalizable overthinking patterns across diverse input datasets, thereby enabling a more principled quantification of overthinking tendencies.

- We introduce an innovative difficulty-aware length penalty and an advantage clipping strategy that jointly prevent distortion in token-level advantage estimation, adaptively reducing response length according to problem difficulty.

## 2 PRELIMINARY

Given a prompt $x = [x_1, \ldots, x_n, <\text{think}>]$, $x = [x_1, \ldots, x_n]$ denotes the user tokens, and $<\text{think}>$ is a special token to trigger the generation of reasoning trajectories (DeepSeek-AI et al., 2025). LRMs generates a response $y = [y_1, \ldots, y_l, </\text{think}>, y_{l+2}, \ldots, y_m]$ where $y = [y_1, \ldots, y_l]$ denotes chains of thought, and $[y_{l+2}, \ldots, y_m]$ represents the summary of the long CoT. Typically, the LRMs produces an excessively long CoT, *i.e.* $[y_1, \ldots, y_l]$, which leads to overthinking and unnecessary reasoning trajectories.

We observe that in the naive GRPO framework, the advantage for each rollout is computed as a single sequence-level value, as shown in Eq.1:

$$\hat{A}_{i,t} = \hat{A}_i = \frac{r_i - \text{mean}(r)}{\text{std}(r)} \tag{1}$$

---

**Algorithm 1 DEPO**: **DE**coupled Advantage **P**olicy **O**ptimization for Efficient Thinking

---

**Input:** Initial policy model $\pi_\theta$, generative reward model GRM, task dataset $\mathcal{D}$, hyperparameters
    $\alpha, \beta$
  1: **for** step $= 1, \ldots, M$ **do**
  2:    Sample a batch $\mathcal{D}_b$ from dataset $\mathcal{D}$
  3:    Sample $G$ outputs $\{o_i\}_{i=1}^{G} \sim \pi_{\theta_{old}}(\cdot|x)$ for each prompt $x \in \mathcal{D}_b$
  4:    Compute accuracy reward $R_{\text{accuracy}}$ (Eq.6) and length reward $R_{\text{length}}$ (Eq.7)
  5:    Compute sequence-level Advantages $\hat{A}'_i$ (Eq.1) and clip biased Advantages $\hat{A}_i$(Eq.9)
  6:    Match redundant reasoning steps in $o_{ie}$ of correct output(Eq.2 ) and compute token-level
       Advantages $\hat{A}_{i,t}$ (Eq.4)
  7:    Update the policy $\pi_\theta$ by maximum $\mathcal{J}_{\text{DEPO}}(\theta)$ (Eq.3)
  8: **end for**
**Output:** $\pi_\theta$

---

Here, $\hat{A}_i$ denotes the normalized advantage of the i-th rollout, and all tokens in the response ($[y_1, \ldots, y_m]$) are assigned this identical advantage value, regardless of whether they belong to efficient or inefficient reasoning segments. This design inherently limits the model's ability to distinguish between efficient and inefficient tokens during optimization. To address this limitation, we refine the original response into $y = [y_1, \ldots, y_{ans}, y_{ans+1}, \ldots, y_l, </\text{think}>, y_{l+2}, \ldots, y_m]$, which explicitly separates the reasoning trajectory into efficient and inefficient segments. Specifically, we define $[y_1, \ldots, y_{ans}]$, which first derives the correct answer as an efficient segment. And we define $[y_{ans+1}, \ldots, y_l]$, which could contain verification or self-reflection to the correct answer $y_{ans}$ as an inefficient segment.

## 3 METHODOLOGY

Our DEPO algorithm comprises three key components: (1) an advantage decoupled computation algorithm for efficient and inefficient tokens, which reduces the update weights of inefficient segments; (2) a difficulty-aware length penalty to reduce the overall response length of models; (3) an advantage clipping strategy designed to mitigate gradient bias in policy optimization induced by the length penalty. By employing these methods, DEPO effectively identifies and suppresses redundant reasoning, *i.e.* $[y_{ans+1}, \ldots, y_l]$, and reduces the overall length of the model's responses without misleading gradient update in policy optimization. The overall algorithmic pipeline is outlined in Algorithm 1.

### 3.1 DECOUPLED ADVANTAGE

We propose an advantage decoupled computation method that guides the model to learn primarily from efficient tokens while suppressing redundant reasoning. Specifically, we denote the tokens after the sentence, first deriving the correct answer, as the inefficient part, *i.e.* $[y_{ans+1}, \ldots, y_l]$. And we fine-tuned a generative reward model (GRM) to accurately identify the token that derives the correct answer ($y_{ans}$) and split off inefficient tokens. Additionally, we substituted GRM for rule to score responses, as GRM significantly outperforms rule-based methods in scoring accuracy. As shown in Fig.2 , we sampled 1024 rollouts from the DeepScaleR (Luo et al. (2025)) dataset using DEEPSEEK-DISTILL-QWEN-7B and categorized them by difficulty as shown in Fig.2.

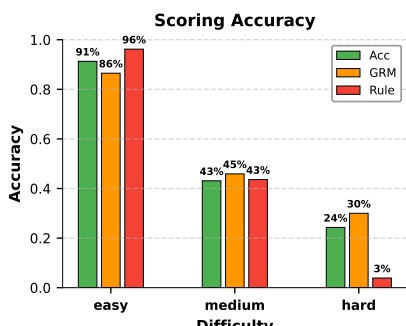

**Figure 2: Scoring accuracy of rule and GRM across difficulty levels.**

The results show that GRM achieves higher scoring accuracy on challenging tasks with complex answer formats. However, this comes at the cost of increased GPU memory consumption. Detailed usage of GRM is provided in Appendix.B.

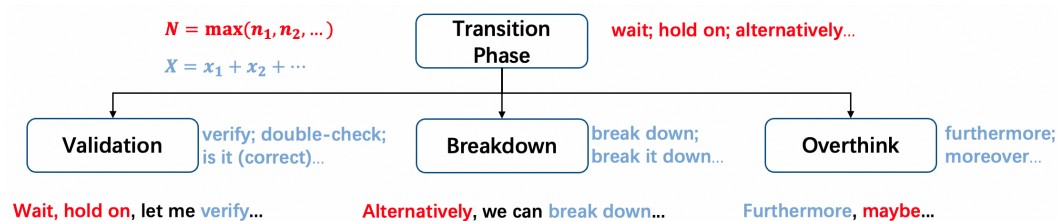

**Figure 3: The redundant reasoning matching method of DEPO**

Additionally, we reduce the advantage value of inefficient tokens based on the number of redundant reasoning paths they contain. To this end, we analyze responses from DEEPSEEK-DISTILL-QWEN-7B to catalog patterns of overthinking behavior, and develop a quantifiable method for identifying overthinking and redundant reasoning steps within the segment $[y_{ans+1}, \ldots, y_l]$, as illustrated in Fig.3, in which $N$ denotes the maximum of transition phrases that start an alternative reasoning step, and $X$ denotes the total number of self-reflection words in $o_{ie}$.

And the redundant reasoning steps in $o_{ie}$ is formulated as:

$$K = \max(N, X) \tag{2}$$

For a given prompt $x$ and generated output $o_i$, the final loss functino of DEPO is:

$$\mathcal{J}_{\text{DEPO}}(\theta) = \mathbb{E}_{x \sim D, \{o_i\}_{i=1}^G \sim \pi_{\theta_{\text{old}}}(\cdot|x)} \left[ \frac{1}{\sum_{i=1}^G |o_i'|} \sum_{i=1}^G \sum_{t=1}^{|o_i'|} \right.$$
$$\left. \left\{ \min\left[ \frac{\pi_\theta}{\pi_{\text{old}}} \cdot \hat{A}_{i,t}, \text{clip}\left(\frac{\pi_\theta}{\pi_{\text{old}}}, 1-\epsilon, 1+\epsilon\right) \cdot \hat{A}_{i,t} \right] \right\} \right] \tag{3}$$

We convert the original sequence-level advantage value $(\hat{A}_i)$ to a token-level advantage value $(\hat{A}_{i,t})$ compared to the objective of naive GRPO. Specifically, we decompose the reasoning process into distinct segments: an efficient segment $o_e = [y_1 \ldots, y_{ans}]$, which directly leads to the correct answer, and an inefficient segment $o_{ie} = [y_{ans+1}, \ldots, </\text{think}>]$, representing overthinking steps. To decouple the contribution of these segments, we lower the advantage values of $o_{ie}$ based on the number of its redundant reasoning steps. The advantage computation method of DEPO is defined as:

$$\hat{A}_{i,t} = \begin{cases} f(o_{ie}) \cdot \hat{A}_i, & \text{if } y_t \text{ in } o_{ie} \text{ and } o_i \text{ is correct} \\ \hat{A}_i, & \text{otherwise} \end{cases} \tag{4}$$

where $\hat{A}_i$ is the sequence-level advantage of response $o_i$ and $f$ is formulated as:

$$f(\cdot) = 1 - \beta \cdot (1 - e^{-\beta \cdot K}) \tag{5}$$

where $\beta$ is hyperparameter and $K$ denotes the number of redundant reasoning steps identified in $o_{ie}$ as presented in Eq.2, through our predefined rule-based method. The range of function $f(\cdot)$ is dependent on $\beta$, and $f(\cdot)$ decreases monotonically with the increase of $K$.

Following the experimental setup of DAPO (Yu et al. (2025)), we compute the final loss across all tokens within a group, while explicitly excluding the KL Divergence term to better enhance models' reasoning capabilities.

### 3.2 DIFFICULTY-AWARE LENGTH PENALTY

In addition to the aforementioned method for reducing redundant reasoning, we also incorporate a length penalty mechanism to minimize the overall response length of the model. The core idea is to reward shorter responses and penalize longer ones within a group of rollouts, particularly for simple questions that require only a minimal number of reasoning tokens. Given a prompt $x$ and a group

of rollouts $\{o_1, \ldots, o_G\}$, we denote their respective lengths as $\{l_1, \ldots, l_G\}$. The accuracy reward is then defined as follows:

$$R_{\text{accuracy}}(o_i \mid x) = \begin{cases} 1, & \text{if } o_i \text{ is correct} \\ 0, & \text{if } o_i \text{ is incorrect} \\ -1, & \text{if } o_i \text{ is overlong} \end{cases} \tag{6}$$

We introduce a negative reward $R_{\text{accuracy}} = -1$ for responses that exceed the maximum allowed response length, treating such cases as more severe failures than merely incorrect answers. This design is motivated by our preliminary experiments, which revealed that approximately 10% of model generations exhibited excessive repetition, resulting in abnormally long outputs.

**Length Penalty.** Furthermore, we introduce a length reward based on the length variance among correct responses and the difficulty of question $x$, and we present the formulation as follows:

$$R_{\text{length}}(o_i \mid x) = \begin{cases} -\alpha \cdot (1 - e^{-\alpha \cdot \delta}) \cdot \frac{|o_i| - \text{mean}(l_{\text{pos}})}{\text{std}(l_{\text{pos}})}, & \text{if } o_i \text{ is correct} \\ 0, & \text{if } o_i \text{ is incorrect} \end{cases} \tag{7}$$

where $\alpha$ is a hyperparameter, $l_{pos}$ denotes the lengths of correct responses, and $\delta$ represents the number of correct responses in a group of rollouts, which means the difficulty of input question $x$. So the final reward of response $o_i$ is:

$$R(o_i \mid x) = R_{\text{accuracy}}(o_i \mid x) + R_{\text{length}}(o_i \mid x) \tag{8}$$

## 3.3 ADVANTAGE CLIPPING

In the computation of advantages compared to naive GRPO, we introduce a clipping operation to the original computation method. As revealed in our preliminary experiments in Fig.4, length-based reward signals, *e.g.* $R_{accuary} = -1$ for excessively long responses and $R_{length}$ for correct responses, could introduce advantage estimation biases: (1) correct responses which are penalized for excessive length may yield negative advantages; (2) incorrect responses could exhibit positive advantages when co-occurring with responses that exceed the maximum length. These cases could mislead policy updates, ultimately causing accuracy degradation. To address this, we propose the following clipping method:

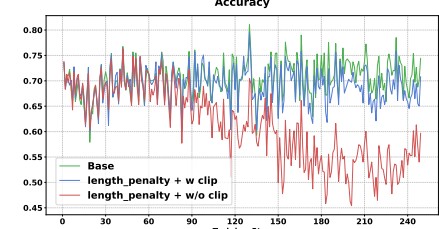

**Figure 4: Training accuracy comparison of naive GRPO, length penalty with *adv_clip* and length penalty without *adv_clip*.**

$$\hat{A}_i = \begin{cases} \text{clip}(\hat{A}'_i, \min(\hat{A}'_{\text{pos}}), +\infty), & \text{if } o_i \text{ is correct} \\ \text{clip}(\hat{A}'_i, -\infty, 0), & \text{if } o_i \text{ is incorrect} \end{cases} \tag{9}$$

where $\hat{A}'_i$ is the original advantage of response $o_i$, and $\min(\hat{A}'_{pos})$ denotes the minimum positive value among unclipped advantage values in $\{o_1, \ldots, o_G\}$. Our method ensures that correct answers consistently yield positive advantages, whereas incorrect answers produce strictly negative values, thereby preventing the model's gradient updates from being trapped in conflicting optimization directions.

## 4 EXPERIMENTS

### 4.1 SETUP

**Model.** We adopt DEEPSEEK-DISTILL-QWEN-7B (DeepSeek-AI et al. (2025)) and DEEPSEEK-DISTILL-QWEN-1.5B (DeepSeek-AI et al. (2025)) as our policy model, which has superior performance in mathematical problems and exhibits twice the length of reasoning compared to its base model.

**Dataset and Metric.** We select DeepScaleR (Luo et al. (2025)) as our training dataset, which consists of approximately 40,000 unique mathematics problem-answer pairs compiled from AIME

1984-2023, AMC (prior to 2023), Omni-Math (Gao et al. (2024)), and STILL (Min et al. (2024)). For the evaluation task, we adopt four math datasets as our test datasets: AMC23, MATH500 (Lightman et al. (2023)), AIME24 and AIME25. Given the limited size of samples in AIME24, AIME25, and AMC23 (30, 30, and 40 instances, respectively), we repeatedly sample each case in these datasets 16 times and adopt the average accuracy (avg@16) as the evaluation metric. For the remaining datasets, *i.e.* MATH500, we uniformly used pass@1 as the metric. The top-p and temperature of the evaluation task are 0.95 and 0.7, and the maximum context size is 16K.

**GRM.** To accurately score the model's responses and extract the first reasoning sentence that leads to the correct answer, we fine-tuned the Qwen2.5-Instruct-7B model (Qwen et al. (2025)) via Supervised Fine-Tuning (SFT) to serve as the GRM. Detailed fine-tuning procedures are provided in Appendix.C.

**Implementation.** We conducted experiments on the VeRL framework (Sheng et al. (2025)). The maximum response length, training batch size and learning rate are set to 16K, 128, 1e-6 respectively, and the hyperparameters of $\alpha$, $\beta$, and the number of rollouts $G$ are 0.2, 0.5 and 8, respectively. We implement alternating execution of the GRM scoring and policy model training tasks through vLLM offload, with the GRM's top-p and temperature parameters set to 0.95 and 1, respectively. The comparative analysis of length penalty versus advantage decoupled computation is detailed in the ablation experiment. Additionally, we use one $8\times$H20 node to train the model for 1 epoch, and we select the checkpoint that achieves an optimal balance between response length and accuracy during training as the baseline for comparison.

## 4.2 BASELINES

We compared the performance of DEPO with the following methods in terms of accuracy and model response length:

- **GRPO** (Shao et al. (2024)) proposes a group-related optimization algorithm that computes sequence-level advantages from a group of rollouts for policy optimization.
- **DAST** (Shen et al. (2025)) constructs preference data by ranking pre-sampled responses using a length-based reward function, and then applies SimPO (Meng et al. (2024)) to fine-tune the model.
- **GRPO-LEAD** (Zhang & Zuo (2025)) introduces a length-dependent accuracy penalty to promote concise generation and an explicit penalty mechanism for incorrect responses, and re-weights advantage values based on problem difficulty, and then fine-tunes the model via GRPO.
- **LC-R1** (Cheng et al. (2025)) first calculates a length reward based on the ratio of the response length to the maximum length within the same group. Then, it extracts valid thinking tokens from the CoT process using the LC-Extractor module, which are treated as the compression part. Finally, it trains the model by computing the loss on the compression and invalid thinking part, respectively.

## 4.3 EXPERIMENT RESULTS

In this section, we present a comprehensive evaluation comparing DEPO with baseline methods across multiple dimensions. As shown in Table.1, for DEEPSEEK-DISTILL-QWEN-7B, DEPO achieves substantial length reduction of 38.3% and 35.9% on challenging problem sets, *i.e.* AIME24 and AIME25, attaining the shortest generated length among all compared baselines. Besides, regarding the accuracy metric, DEPO shows stable performance compared to the original model. With accuracy fluctuations ranging from -0.6% regression and +3.1% improvement across challenging datasets. However, DEPO incurs approximately 3% loss of average accuracy versus naive GRPO on challenging datasets, as its aggressive length optimization may compromise complex reasoning steps required for these tasks. Furthermore, in simple datasets (AMC23, MATH500), DEPO achieves the highest accuracy while maintaining second-best length efficiency, demonstrating optimal performance for routine tasks requiring both precision and conciseness. For the smaller DEEPSEEK-DISTILL-QWEN-1.5B model, DEPO reduces response length by 39.1% while achieving an accuracy that is only 0.2% lower than GRPO and 2.1% higher than the base model, effectively improving accuracy while substantially shortening model responses.

Table 1: **Accuracy (Acc) and response length (Length) of different methods on AIME24, AIME25, AMC23, MATH500. The best and second results are bold and underlined respectively.**

| Method | AIME24 | | AIME25 | | AMC23 | | MATH500 | | Avg | |
|---|---|---|---|---|---|---|---|---|---|---|
| | Acc | Length | Acc | Length | Acc | Length | Acc | Length | Acc | Length |
| DeepSeek-R1-Distill-Qwen-7B | | | | | | | | | | |
| Original | 49.6 | 10670 | 39.8 | 11068 | 87.8 | 5794 | 92.8 | 3601 | 69.3 | 7591 |
| GRPO | **55.0** | 9913 | **42.0** | 10709 | 88.1 | 5613 | 93.4 | 3527 | **71.3** | 7263 |
| DAST | 52.9 | 9674 | 36.0 | 10729 | 88.8 | 5091 | 93.6 | 2904 | 69.7 | 6906 |
| LC_R1 | 49.2 | 7013 | 37.3 | 7530 | 87.3 | **2847** | 93.4 | **2276** | 68.6 | 4733 |
| GRPO-Lead | 49.2 | 9507 | 39.2 | 9529 | 89.0 | 4525 | 93.8 | 2957 | 69.7 | 6434 |
| DEPO | 52.7 | **6580** | 39.2 | **7092** | **90.5** | 3215 | **94.4** | 2318 | 71.1 | **4656** |
| DeepSeek-R1-Distill-Qwen-1.5B | | | | | | | | | | |
| Original | 30.2 | 12165 | 24.4 | 12109 | 70.0 | 7720 | 84.6 | 4820 | 54.0 | 9048 |
| GRPO | **33.5** | 10609 | **27.3** | 10668 | **74.4** | 6561 | 86.8 | 4208 | **57.2** | 7864 |
| DEPO | 30.8 | **7732** | 24.8 | **7649** | 74.2 | **4388** | **87.2** | **2762** | 56.1 | **5510** |

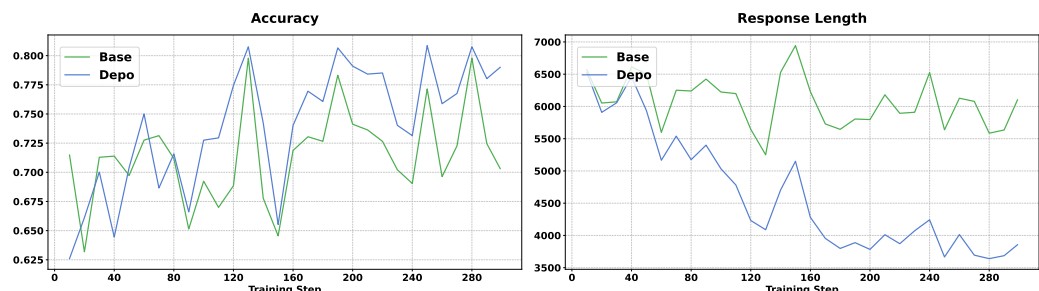

Figure 5: **Comparison of model accuracy and response length on DeepScaleR between naive GRPO and DEPO at different training steps.**

During the training process on DEEPSEEK-DISTILL-QWEN-7B as shown in Fig.5, although the scoring accuracy of GRM and the rule-based method shows minor discrepancy (3-5% divergence), their overall trends of scoring remain aligned. Besides, DEPO significantly reduces model response length during training, decreasing the average from approximately 6,500 tokens to around 3,600 tokens, which represents a 44% reduction in sequence length.

**DEPO suppresses repetitive outputs and redundant self-reflection compared to naive GRPO.** As shown in Fig.6, our experimental results demonstrate that DEPO effectively mitigates repetition-induced overlong responses during generation. Compared to baseline GRPO training—which produces approximately 110 overlong outputs per 1024 rollout (accounting for 10.7% of total generations), DEPO reduces this frequency to just 1-2 occurrences (0.1% of rollouts), representing a 98% decrease in overlong outputs. The phenomenon observed in these cases primarily arise from two key factors: first, repetitive verification—where the model's self-reflection behavior leads to redundant validation in its reasoning process (e.g., repeated phrases like "let's check again" or "wait, hold on"), often resulting in excessively long responses; second, auto regressive error accumulation in LRMs, where gradual error buildup during generation causes the probability of certain tokens to incrementally increase, leading to unintended repetition of those tokens in the sequence. DEPO effectively mitigates this issue by imposing a stricter penalty on repetitive patterns, directly addressing both the reflective loops from self-reflection and the token-level redundancy from auto regressive errors.

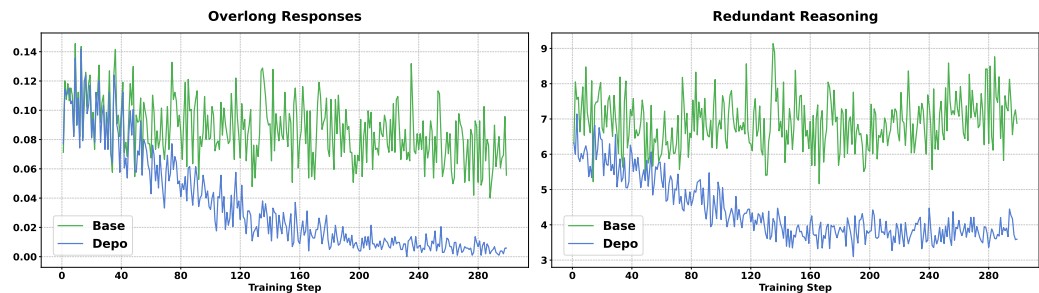

**Figure 6:** Comparison of overlong responses ratios and redundant reasoning steps in inefficient segments per rollout sample between naive GRPO and DEPO.

**DEPO Reduces Redundant Reasoning in Inefficient Segments.** As shown in the right panel of Fig.6, DEPO significantly reduces redundant reasoning steps—where the model keeps re-verifying a correct answer—by using a rule-based matching mechanism (illustrated in Fig.3). This approach reduces redundant verification by about half compared to GRPO, significantly improving the efficiency of the model's reasoning process.

## 4.4 ABLATION RESULTS

We evaluate the contribution of each component of DEPO, *i.e.* Adv_Decouple and Len_Penalty, through an ablation experiment on DEEPSEEK-DISTILL-QWEN-7B. We present the ablation results in Table.2

**Table 2:** Ablation Study of Advantage Decoupling (Adv-Decouple) and Length Penalty (Len-Penalty) on Model Accuracy and Redundant Reasoning

| Method | AIME24 Acc Len | AIME24 Reflect | AIME25 Acc Len | AIME25 Reflect | AMC23 Acc Len | AMC23 Reflect | MATH500 Acc Len | MATH500 Reflect | Avg Acc Len | Avg Reflect |
|---|---|---|---|---|---|---|---|---|---|---|
| DEPO | **52.7** **6580** | 5.2 | **39.2** **7092** | 5.5 | **90.5** **3215** | **2.1** | 94.4 **2318** | **1.6** | **71.1** **4656** | 3.5 |
| -w/o Adv-Decouple | 50.4 7002 | 6.8 | 37.7 7300 | 7.3 | 86.6 3450 | 2.8 | 93.8 2620 | 3.1 | 68.9 4944 | 4.8 |
| -w/o Len-Penalty | 52.1 6962 | **5.1** | 38.8 7638 | **5.2** | 88.4 3721 | **2.1** | **94.8** 2953 | 1.7 | 70.3 5174 | **3.4** |

**The accuracy and length trade-off of length penalty.** As shown in Table.2, the ablation variant without advantage decoupling (w/o Adv-Decouple)—which relies solely on length penalty—produces consistently shorter responses than the variant without length penalty (w/o Len-Penalty). This indicates that the length penalty is more effective at reducing output length than our advantage decoupling mechanism alone, yielding an average additional reduction of about 300 tokens across most datasets. However, this comes at a slight cost in accuracy: even with advantage clipping (Sec.3.3), the length-penalty-only model underperforms the advantage decoupling model. These results highlight the need to carefully balance response length and accuracy when applying length penalties.

**Advantage decoupling versus length penalty in suppressing self-reflection.** We further quantify the number of redundant reasoning steps (e.g., "double-check" or "wait, hold on") in the inefficient reasoning segments ($o_{ie}$) across ablation settings. The results show that advantage decoupling reduces such redundant behaviors more effectively than the length penalty, leading to fewer unnec-

essary verification steps or shifts to alternative reasoning paths after the model has already reached the correct answer. Moreover, in the full DEPO model, advantage decoupling remains the dominant component for both suppressing self-reflection and improving accuracy—outperforming length penalty in these aspects.

## 5 RELATED WORK

**Reasoning of CoT in LRMs.** Following Wei et al. (2023)'s demonstration that extended Chain-of-Thought (CoT) enhances Large Reasoning Models (LRMs), frontier models like OpenAI o1 (OpenAI (2024)) and DeepSeek-R1 (DeepSeek-AI et al. (2025)) now employ reinforcement learning to fine-tune reasoning trajectories. During this process, models dynamically verify and switch reasoning paths—termed *aha-moment* (DeepSeek-AI et al. (2025))—when encountering solution uncertainty. However, persistent *aha-moments* after correct answers cause excessive verification of already-correct solutions and lead to redundant, lengthy responses, particularly detrimental in mathematical and coding benchmarks, which we denote as *overthink* (Cuadron et al. (2025)).

**Efficient Reasoning for LRMs.** Recent methods for improving reasoning efficiency and mitigating overthinking in LRMs typically aim to reduce output tokens. This is achieved through reward shaping based on response length, either by rewarding shorter rollouts during training (Arora & Zanette (2025)) or setting a "best-length" threshold (Liu et al. (2025)). Alternatively, other approaches include fine-tuning on length preference pairs (Shen et al. (2025))or employing prompt engineering to elicit shorter responses (Han et al. (2025)). More novel techniques encourage models to dynamically decide whether to use Chain-of-Thought (CoT) based on problem difficulty (Zhang et al. (2025)), or extract only valid reasoning segments for training via auxiliary modules (Cheng et al. (2025)). However, these works mentioned treat model output as a whole during training, failing to distinguish between efficient and inefficient reasoning segments or evaluate the precise self-reflection mechanisms of overthinking. Motivated by this, we propose DEPO in our work, which reduces output length and redundant self-reflection by decoupling their advantage computations for efficient and inefficient reasoning components according to the degree of overthinking, which is a novel direction for efficient reasoning.

## 6 LIMITATION

In this section, we discuss several limitations of our work: (1) Our training is confined to mathematical datasets since they are easy to verify, and the model's responses contain explicit reasoning steps that derive the correct answer. Further studies are needed to evaluate the effectiveness of DEPO on other domains such as logical and code problems. (2) Due to limitations in computational resources and the variance of overthinking between different models, we only conducted experiments on DEEPSEEK-DISTILL-QWEN-7B and DEEPSEEK-DISTILL-QWEN-1.5B. Nevertheless, DEPO still demonstrated its efficacy for reducing response length and redundant reflection in reasoning steps. Besides, since DEPO relies on GRM for both scoring and identifying the first correct reasoning step; thus, its performance critically depends on GRM quality. We address this by rigorously filtering the GRM training data, with details provided in Appendix.C.

## 7 CONCLUSION

In this paper, we propose DEPO, a reinforcement learning algorithm designed to mitigate overthinking in LRMs. DEPO addresses this issue through two core mechanisms: (1) it decouples the loss computation for inefficient reasoning segments from the rest of the response, enabling the model to explicitly learn to suppress redundant tokens; (2) it incorporates a length penalty into the reward function, encouraging the model to generate shorter outputs. Extensive experiments demonstrate that DEPO effectively balances accuracy and response length, while significantly reducing redundant reasoning steps across diverse mathematical problem sets.

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
