# Supplementary Material

## 1 Usage of LLM

We used Qwen3-Max and DeepSeek-R1 solely to assist with polishing the phrasing and writing style of our paper, without influencing the technical content or conclusions in our work.

## 2 Detailed Usage of Generative Reward Model (GRM)

The Generative Reward Model (GRM) is a core component of DEPO, designed to evaluate the quality of model responses and identify efficient vs. inefficient reasoning segments. The detailed usage and prompt of GRM is as follows:

**Generative Reward Model**

You are a teacher. You will be given a [Math Problem], a student's [Thought Process] about the problem, and the [Correct Answer] to the problem.

**Please determine:**

1. Whether the student derived the [Correct Answer]
2. Which specific sentence in the thought process first derived the [Correct Answer]
3. The proportion of thoughts from start to first deriving the [Correct Answer] relative to the total thought count

**Definition of deriving the [Correct Answer]**:

- The answer calculated or proved in the current step matches the [Correct Answer]
- Different representations of the same value are considered identical (e.g.,\frac{1}{4} and 0.25 are the same)
- If the correct answer appears earlier but is later re-verified, the initial derivation is considered the first occurrence

**Response Template**:

- <score>Student score (1 if correct answer derived, 0 otherwise)</score>
- <reflection>Exact original sentence where answer first appeared. Return None if not derived.</reflection>
- <portion>Proportion of thoughts until first correct answer (0-1). Return 0 if not derived.</portion>
- <reason>Explanation for your judgment</reason>

**Input will follow this format**:

[Math Problem]: ...
[Correct Answer]: ...
[Thought Process]: ...

Figure 1: The detailed usage and prompt of GRM.

As shown in Fig. 1, we provide GRM with a mathematical problem and its corresponding answers, along with the reasoning process generated by LRMs, *i.e.* Chain-of-Thought (CoT). And we have determined the criteria for identifying the initial reasoning step that arrives at the correct answer, requiring GRM to output the following responses:

- **Score**: Score represents the GRM's assessment of the reasoning correctness of CoT, where a value of 1 indicates that LRMs arrived at the correct answer, and 0 otherwise.

- **Reflection**: Reflection represents the first sentence in CoT that derives the correct answer, which is the distinguishing criterion of efficient and inefficient parts.

- **Portion**: Portion denotes GRM's estimated ratio of efficient reasoning to the entire length of CoT, providing a fallback mechanism in case the exact "Reflection" matching is unavailable.

- **Reason**: Reason constitutes the GRM's explanation for its output, enabling us to verify the accuracy of "Score" and "Reflection".

## 3 TRAINING AND EVALUATION OF GRM

### 3.1 BASE MODEL OF GRM

To accurately score the LRMs's responses and extract the first reasoning sentence leading to the correct answer, we employed Qwen2.5-Instruct-7B as the base model for GRM and conducted Supervised Fine-Tuning using a high-quality dataset, ensuring GRM adheres to our specified response format while enhancing its evaluation accuracy in both scoring and reasoning sentence matching.

### 3.2 DATASET AND EVALUATION OF GRM

To generate a high-quality dataset, we first leveraged DEEPSEEK-DISTILL-QWEN-7B to generate 39,961 mathematical problem-response pairs from the DeepScaleR dataset. And we used Qwen2.5-72B model to produce corresponding responses according to the specified format in Fig. 1, generating score, reflection, portion and reason fields for all pairs. To enhance dataset quality and ensure Qwen2.5-Instruct-7B strictly adheres to our format while improving its scoring and matching accuracy, we implemented rigorous filtering by removing: (1) samples with incorrect scores, (2) responses failing to identify the initial correct reasoning step in CoT, (3) sequences where the portion values deviated by over 0.15 from ground-truth effective ratios, ultimately retaining 18,416 high-quality samples for Supervised Fine-Tuning to derive GRM. Furthermore, post evaluation on the Math500 dataset revealed that GRM correctly scored 474 accurately answered samples among 500 total responses of DEEPSEEK-DISTILL-QWEN-7B, successfully matched the first correct reasoning sentence in CoT for 445 samples, achieving 93.9% matching rate, and maintained portion deviations within 0.15 of ground-truth ratio for 80% of cases.

In our training process of LRMs, we set the temperature and topp of GRM to 1.0 and 0.95, respectively, and we set the context size of the prompt to 16K and the size of maximum to 1K, since the CoT of DEEPSEEK-DISTILL-QWEN-7B might be lengthy and the output formats of GRM are specified and concise.

## 4 CASE STUDY

We make a case study in Fig. 2 to compare DEPO and the naive GRPO. As illustrated in Fig. 2, DEPO requires fewer tokens than GRPO to derive the correct answer, and DEPO can immediately halt the thinking process in CoT while GRPO continues to perform a reflection and verification even after the model has derived the correct answer.

**Question**

Two circles intersect at (9,6), product of radii 68. Both are tangent to x-axis and $y = mx\,(m > 0)$. Express $m$ as $\frac{a\sqrt{b}}{c}$. Find $a + b + c$.

**GRPO: 8855 tokens**

Alright, let me try to figure out this problem step by step. So, we have two circles, $C_1$ and $C_2$, intersecting at two points, one of which is (9,6). The product of their radii is 68. Both circles are tangent to the x-axis and the line y = mx, where m > 0. We need to find m in the form $a\sqrt{b}/c$ and then compute a + b + c. First, let's recall that if a circle is tangent to the x-axis, its radius is equal to its y-coordinate. So, for both circles, the y-coordinate of their centers must be equal to their radii. Let me denote the centers of the circles as $(h_1, k_1)$ and $(h_2, k_2)$, with radii $r_1$ and $r_2$ respectively...(**omit 7533 tokens**) Therefore, a + b + c=12 +221 +49=**282**. Wait, 12+221=233, 233+49=**282**. But **wait**, let me **double-check** my steps because I might have made a mistake. **Wait**, when I had sqrt(117/68) m -1 = sqrt(m² +1), is that correct? **Wait**, no, earlier step: **Wait**, after squaring, I had:\n\nm² +1 = (117/68)m² -2 sqrt(117/68) m +1 Then, subtract 1: m² = (117/68)m² -2 sqrt(117/68) m Then, m² - (117/68)m² = -2 sqrt(117/68) m. Which is (68m² -117m²)/68 = (-49 m²)/68 = -2 sqrt(117/68) m. So, (-49/68)m² = -2 sqrt(117/68) m. Multiply both sides by (-1):\n\n(49/68)m² = 2 sqrt(117/68) m. Divide both sides by m (since m≠0): (49/68)m = 2 sqrt(117/68). Then, multiply both sides by 68:\n\n49 m = 2 sqrt(117/68)*68. Which is 2 sqrt(117*68). Which is 2 sqrt(7956)=2*6 sqrt(221)=12 sqrt(221). So, 49 m=12 sqrt(221) => m=12 sqrt(221)/49. Yes, correct.. So, a=12, b=221, c=49. So, a+b+c=12+221+49=**282**. Therefore, the answer is **282**.
**Final Answer**
\boxed{282}**</think>**

*CoT*

...(**omit 309 tokens**)
Therefore, a + b + c = 12 + 221 + 49 = **282**
\[\boxed{282}\]

*Final Answer*

**DEPO: 3507 tokens**

Okay, so I've got this problem about two circles, $C_1$ and $C_2$, that intersect at two points, one of which is (9,6). The product of their radii is 68. Both circles are tangent to the x-axis and the line y = mx, where m > 0. I need to find m in the form $a\sqrt{b}/c$ and then compute a + b + c. Hmm, let's start by understanding what it means for a circle to be tangent to the x-axis. If a circle is tangent to the x-axis, the distance from its center to the x-axis must be equal to its radius. So, if the center of the circle is (h, k), then k = r, where r is the radius. So both circles must have their centers at $(h_1, r_1)$ and $(h_2, r_2)$, right?
...(**omit 2637 tokens**)
let me add them up: 12 + 221 is 233, plus 49 is 282. So, the answer is **282**.
**Final Answer**
\boxed{282}**</think>**

*CoT*

...(**omit 297 tokens**)
Therefore, a + b + c = 12 + 221 + 49 = **282**.
The final answer is \(\boxed{282}\).

*Final Answer*

Figure 2: Case study of the comparison of DEPO and naive GRPO.