# OpenReview forum: "Towards Flash Thinking via Decoupled Advantage Policy Optimization"
_ICLR.cc/2026/Conference — ICLR 2026 Conference Withdrawn Submission_

### Official Review · Reviewer_2Axb · 2025-10-20

**Soundness:** 3
**Presentation:** 3
**Contribution:** 2
**Rating:** 4
**Confidence:** 4

**Summary:**

This work addresses the overthinking problem of LLMs in tasks such as mathematics and proposes DEPO. It introduces three key designs to shorten responses and suppress redundant reasoning: (1) Decoupled advantage estimation: tokens before the correct answer are regarded as efficient segments, while those after it are inefficient segments, whose advantages are down-weighted according to their redundancy;(2) Difficulty-aware length penalty: based on group-wise rolling sampling, overly long or verbose outputs for easy samples are penalized; (3) Advantage clipping: prevents sign reversal in the advantage term caused by length rewards, ensuring stable policy updates. The method relies on a GRM that automatically identifies the token where the correct answer first appears, enabling the segmentation of efficient and inefficient parts. Experiments on several datasets demonstrate its effectiveness.

**Strengths:**

- This work focuses on the overthinking problem of large language models (LLMs), which leads to increased latency and computational cost.

- It proposes DEPO, which refines sequence-level advantage estimation to the token level. The advantages of inefficient segments are monotonically down-weighted according to the number of redundant steps, offering finer granularity than methods that rely solely on overall sequence length.

- Experiments on multiple benchmarks demonstrate that the proposed method effectively shortens output length.

**Weaknesses:**

- In real reasoning scenarios, verification or revision steps after the first correct one may enhance robustness or reduce hallucinations, and thus may not always be penalized. The current setup assumes these tokens are inefficient; are there any counterexamples or boundary analysis. Are there essential reasoning steps still required after the first correct token? Could empirical comparisons be provided to show how removing these later steps affects accuracy?

- The overthinking issue has been widely discussed in prior work, e.g., [1–3], which improves reasoning accuracy by maximizing per-token utility and reducing redundancy. In contrast, manually segmenting tokens and applying post-training strategies with length penalties may cause performance regression (e.g., there exists a 3% average accuracy drop of DEPO in Table 1). Further clarification of this work’s advantages over existing approaches is necessary. [1] Do not think that much for 2 + 3 =? On the overthinking of o1-like LLMs. [2] Optimizing Test-Time Compute via Meta Reinforcement Fine-Tuning. [3] Learning to Think: Information-Theoretic Reinforcement Fine-Tuning for LLMs.

- Although the paper acknowledges the high memory cost and better performance on hard problems of GRM, its errors may directly affect segmentation and advantage estimation. Has the sensitivity of DEPO to GRM mispredictions been evaluated? Are there weakly supervised or model-free proxy alternatives? Since K = max(N, X) relies on matching transition or reflexive phrases, can this rule generalize across domains or languages?

- The effect of advantage clipping on estimation bias and variance is preferred, e.g., convergence or stability comparisons in a synthetic environment with a known optimal policy; Some (e.g., LC-R1 and DAST) depend on preference or extraction modules, reporting unified tuning budgets and compute statistics (GPU days, memory usage), along with rerun variance and statistical significance, is recommended.

**Questions:**

Please see Weaknesses.

---

### Official Review · Reviewer_LU5j · 2025-10-26

**Soundness:** 1
**Presentation:** 1
**Contribution:** 1
**Rating:** 2
**Confidence:** 4

**Summary:**

Decoupled Advantage Policy Optimization (DEPO) seeks to reduce the **overthinking** phenomenon in Large Reasoning Models (LRM) by first identifying and decoupling the "efficient" and "inefficient" segments in reasoning traces. Following that, they propose a difficulty-aware length penalty and an advantage clipping strategy to ensure non-distorted advantage estimation in downstream GRPO-like policy optimization. Experiments on math datasets demonstrate that DEPO achieves good reasoning length reduction without compromising accuracies on test math datasets.

**Strengths:**

- The paper makes good observation that existing efficient reasoning (RL) methods fail to do proper credit assignment, treating the whole sequence equally and assigning the same reward. And it thus proposes an interesting "decoupling" idea to identify and separate efficient & inefficient segments in reasoning traces.
- It performs reasonable experiments that illustrate the effectiveness of DEPO on in-domain math problems and the `Deepseek-Distill-Qwen` model series.

**Weaknesses:**

Overall, the biggest issue of this paper is that **the paper is severely incomplete and underdeveloped**: the **Appendix is completely missing** so a lot of important justifications of the method (e.g. how it uses the GRM to select the truncation point) are lacking. The writing also contains typos and the math formulation is difficulty to follow. Given its current state, the paper might be more appropriate for a future submission once these foundational issues are addressed.

### Key Weaknesses

- **Unjustified assumptions about reasoning structure.**
  While the paper acknowledges that reasoning traces contain both efficient and inefficient segments, it builds upon a highly naive assumption that the model *always outputs the efficient portion first*, followed by redundant or unnecessary reasoning that can be truncated. This is a strong and opinionated claim that requires empirical or theoretical justification. In fact, many prior works have observed the opposite pattern -- where overthinking manifests *before* the correct reasoning emerges -- which seems to be ignored based on this paper's assumption.

- **Flawed reward design and post hoc patching via advantage clipping.**
  The paper’s proposed length reward (Eq. 6) assigns **–1** to overlong rollouts but **0** to incorrect ones, leading to pathological behavior where incorrect reasoning may be rewarded more favorably than correct but lengthy reasoning. The introduced “advantage clipping” mechanism appears to be a *post-hoc solution* to this self-inflicted problem. If the authors are already aware that “correct responses penalized for excessive length may yield negative advantages,” then the natural fix should be to **redesign the reward function**, not to introduce additional layers of complexity. As reference, existing works such as [L1](https://arxiv.org/pdf/2503.04697), [O1-Pruner](https://arxiv.org/abs/2501.12570) and [ShorterBetter](https://arxiv.org/pdf/2504.21370) already provide balanced reward formulations that achieve this goal more cleanly.

- **Limited experimental scope.**
   All experiments are conducted solely on `Deepseek-Distilled-Qwen` models and math datasets. Given the well-known concerns surrounding the `Qwen` + Math setup -- such as potential data leakage or mid-training exposure -- it is difficult to attribute the observed improvements to the proposed method rather than dataset or model artifacts. Evaluations on *non-Qwen* models (e.g., Llama family) and *out-of-domain* tasks (e.g., coding or general reasoning benchmarks) are essential to establish the robustness and generality of the approach.

**Questions:**

See the _Weaknesses_ section.

---

### Official Review · Reviewer_wPPm · 2025-10-28

**Soundness:** 2
**Presentation:** 2
**Contribution:** 2
**Rating:** 4
**Confidence:** 5

**Summary:**

This paper addresses the issue of overthinking in Large Reasoning Models, where models generate excessively long and redundant reasoning steps. The authors propose DEPO, a reinforcement learning framework to encourage more concise outputs. The key idea is to decouple the advantage computation. A generative reward model identifies the point where a correct answer is first reached, splitting the response into efficient and inefficient segments. The advantage for tokens in the inefficient segment is then selectively down weighted. Combined with a difficulty aware length penalty and advantage clipping, DEPO significantly reduces response length on mathematical reasoning tasks while maintaining comparable or slightly improved accuracy.

**Strengths:**

1. The core contribution, the decoupled advantage mechanism, is intuitive and well-motivated. By isolating and penalizing only the reasoning steps that occur after a correct solution is found, the method provides a more targeted approach to reducing redundancy compared to global length penalties.

2. The paper presents strong empirical results. The proposed DEPO method achieves a substantial reduction in response length on multiple mathematical reasoning benchmarks, while largely preserving or even slightly improving task accuracy. This demonstrates the practical effectiveness of the approach.

**Weaknesses:**

1. The novelty of decoupling advantage computation appears incremental. The concept of identifying and down-weighting inefficient parts of a reasoning chain shares conceptual similarities with prior work like LC-R1, which also separates valid and invalid tokens.

2. The method's performance is heavily reliant on the accuracy of the Generative Reward Model used for segmentation. The paper does not adequately analyze the impact of GRM errors on the final policy, especially on hard problems where GRM accuracy is lower.

3. The quantification of redundant reasoning relies on a heuristic, rule-based matching of predefined phrases shown in Figure 3. This approach may lack robustness and generalizability, as it is tailored to specific linguistic patterns of overthinking.

**Questions:**

1. Could the authors please elaborate on the key conceptual differences between your decoupled advantage method and previous approaches, such as LC-R1?

2. The method's success depends on the GRM accurately identifying the first correct reasoning step. What is the effect on the learning process when the GRM makes a mistake, for example, by identifying the correct step too early or too late? Have the authors conducted any analysis on the policy model's sensitivity to GRM accuracy?

3. The scaling factor for penalizing inefficient tokens is based on a rule-based system for counting redundant phrases. How sensitive is the proposed method to this specific set of rules?

---

### Official Review · Reviewer_W2rF · 2025-10-31

**Soundness:** 2
**Presentation:** 1
**Contribution:** 1
**Rating:** 2
**Confidence:** 2

**Summary:**

The paper introduces **DEPO (Decoupled Advantage Policy Optimization)**, a variant of GRPO designed to reduce “overthinking” in reasoning-oriented LLMs. DEPO separates tokens into *efficient* and *inefficient* regions using a generative reward model, down-weights advantages in inefficient regions, applies a difficulty-aware length penalty, and clips advantages. Experiments on DeepSeek-Distill-Qwen-7B and 1.5B show approximately 40% shorter outputs while maintaining comparable accuracy to GRPO.

**Strengths:**

- The paper addresses a practical problem, namely excessive verbosity in chain-of-thought reasoning for RL-tuned LLMs.
- The proposed components, including token-level weighting, length-aware penalties, and advantage clipping, are simple and could be easily integrated into existing RL pipelines.
- Experiments consistently demonstrate output shortening effects.

**Weaknesses:**

- The presentation is disorganised and lacks rigour, with key equations and algorithmic steps being unclear.
- The method is largely heuristic, combining known techniques without strong theoretical or empirical justification.
- Dependence on the generative reward model introduces variability/uncertainty, but its reliability and calibration are not evaluated.
- Several mathematical components are ambiguous, including the likelihood ratio in Eq. 3, the function $f(\cdot)$ in Eq. 5, and the relationship between token- and sequence-level advantages (Eq. 4 vs. Eq. 9).
- The length penalty is applied only to correct outputs, which seems arbitrary and could bias training.
- Experimental reporting is incomplete: details on seeds, variance, and hyperparameter sensitivity are missing.
- The study is limited to two closely related models, and generalisation to other architectures is untested.
- Several baselines are missing on the 1.5B model, weakening the comparative analysis.

**Questions:**

1. How many seeds or independent runs were used, and can you report mean ± standard deviation or confidence intervals?
2. Why are baselines such as DAST or LC-R1 missing for the 1.5B model?
3. Does DEPO generalise to other base models such as LLaMA or Qwen2?
4. In Eq. 3, is the likelihood ratio computed per-token or over the entire sequence?
5. In Eq. 5, is $f(\cdot)$ constant across tokens, and how sensitive are results to the value of β?
6. Why is the length penalty applied only to correct outputs?
7. How were $\alpha$, $\beta$, and G chosen, and how robust are results to these settings?
8. In Eq. 9, is clipping applied before or after token-level decoupling?

---

### Official Review · Reviewer_Dcn6 · 2025-11-01

**Soundness:** 3
**Presentation:** 3
**Contribution:** 3
**Rating:** 4
**Confidence:** 4

**Summary:**

The authors present DEPO, a new RL training algorithms that reduces overthinking in large reasoning models. The design includes classifying the responses into efficient and inefficient segments, decoupling token-level advantages for these segments, adding a difficulty-aware length penalty, and introducing advantage clipping to avoid misleading gradient updates. Experiments on math reasoning datasets show that it can achieve comparable performance while significantly reduces the response lengths. The ablation studies examine the effectiveness of all the components in the design.

**Strengths:**

Making RL training inference-aware is an interesting point. Since long response lengths will significantly increase the costs of deployment, thus having a new RL algorithm that can make the reasoning more concise while still have the comparable performance is meaningful. In general the authors' design is sound and tackles quite a few limitations in the previous works.

**Weaknesses:**

I have some questions about the costs of the training and also the generalization of the method.
- A 7B model was used as the GRM while the training was conducted on 1.5B / 7B model. Can you discuss the overhead of having a separate reward model during training? How will the training efficiency be affected?
- Like the authors have mentioned, they only test on math datasets as they are easy to verify. While for other tasks, can we still easily figure out the 'answer tokens' in the responses? For example, for code problems the model may write different versions of codes, how do we know which segment corresponds to the efficient segment? If we are not able to identify these segment, the generalization of this method could be a problem.
- The authors seem to assume that once the model outputs the correct answer in the response, the later parts belong to inefficient segments. While it may make sense for the math problems used in the evaluation, for other reasoning tasks, punishing further reflections may hurt the model performance. Also, is it possible that the efficient and inefficient segments can appear alternately in the response?
- The experiments in the evaluation are only trained for 1 epochs (< 300 steps), which is short. I am not sure whether the training can still be stable with more training steps.

**Questions:**

Please check my questions in the weakness section.

---

### Note · Authors · 2025-11-17

I have read and agree with the venue's withdrawal policy on behalf of myself and my co-authors.